# A Mixed Methods, Quasi-Experimental Evaluation Exploring the Impact of a Secondary School Universal Free School Meals Intervention Pilot

**DOI:** 10.3390/ijerph20065216

**Published:** 2023-03-22

**Authors:** Victoria R. Carlisle, Patricia E. Jessiman, Katie Breheny, Rona Campbell, Russell Jago, Naomi Leonard, Marcus Robinson, Steve Strong, Judi Kidger

**Affiliations:** 1Department of Population Health Sciences, Bristol Medical School, University of Bristol, Bristol BS8 1QU, UK; 2NIHR PHIRST Insight, Bristol Medical School, University of Bristol, Bristol BS8 1QU, UK; 3Centre for Exercise, Nutrition & Health Sciences, School for Policy Studies, University of Bristol, Bristol BS8 1QU, UK; 4The National Institute for Health Research, Applied Research Collaboration West (NIHR ARC West), University Hospitals Bristol and Weston NHS Foundation Trust, Bristol BS1 2NT, UK; 5London Borough of Hammersmith and Fulham, London W6 9JU, UK

**Keywords:** food security, food poverty, universal free school meals, school meals, evaluation, mixed methods

## Abstract

Food insecurity amongst households with children is a growing concern globally. The impacts in children include poor mental health and reduced educational attainment. Providing universal free school meals is one potential way of addressing these impacts. This paper reports findings on the impact of a universal free school meals pilot in two English secondary schools. We adopted a mixed-methods, quasi-experimental design. The intervention schools were one mainstream school (*n* = 414) and one school for students with special educational needs (*n* = 105). Two other schools were used as comparators (*n* = 619; *n* = 117). The data collection comprised a cross sectional student survey during the pilot (*n* = 404); qualitative interviews with students (*n* = 28), parents (*n* = 20) and school staff (*n* = 12); and student observations of lunchtimes (*n* = 57). Qualitative data were analysed using thematic analysis, and descriptive analyses and logistic regressions were conducted on the quantitative data. Self-reports of food insecurity were high at both intervention (26.6%) and comparator schools (25.8%). No effects of the intervention were seen in the quantitative findings on either hunger or food insecurity. Qualitative findings indicated that students, families and staff perceived positive impact on a range of outcomes including food insecurity, hunger, school performance, family stress and a reduction in stigma associated with means-tested free school meals. Our research provides promising evidence in support of universal free school meals in secondary schools as a strategy for addressing growing food insecurity. Future research should robustly test the impact of universal free school meals in a larger sample of secondary schools, using before and after measures as well as a comparator group.

## 1. Introduction

It is currently estimated that 15 percent of households in the UK with children experience food insecurity [1]; in the USA, the figure is 6.2 percent [2]. The UK Trussell Trust define household food insecurity as “a household level economic and social condition of limited or uncertain access to adequate food” and hunger as “a range of experiences falling under severe or moderate household food insecurity” [3]. Household food insecurity disproportionately impacts people with disabilities, single parents and non-white ethnic groups [1,4,5]. The impacts of food insecurity on children are wide ranging, but include poor mental health and psychological distress [6], as well as reduced educational attainment [7,8]. Children who experience food insecurity are more likely to consume foods that are higher in fat and sugar, eat less vegetables [9] and experience physical health complaints that impede school attendance [10]. 

The expansion of free school meals (FSM) provision is one of a range of solutions proposed to address household food insecurity [1]. Currently, Sweden and Finland are the only countries to provide universal free school meals (UFSM) to all age groups; however, studies have taken place globally to explore the potential impacts of UFSM. In the USA, Community Eligibility Provision allows schools where at least 40% of pupils are eligible for FSM to access government funding to provide a free lunch (and breakfast) to all pupils. A recent review of the evidence in relation to Community Eligibility Provision showed an increase in uptake of school meals, as well as improvements in weight outcomes, food insecurity levels and lower rates of disciplinary referrals. Evidence for impacts on test scores and attendance was weaker [11]. In Japan, where school meals have been compulsory (although not free to all) since 1947, there is evidence for the effectiveness in partially reducing socio-economic differences in fruit and vegetable intake [12]. A recent systematic review explored the impacts of UFSM provision (breakfast and lunch) on uptake, child diet, attendance, attainment and body mass index. The review included 47 global studies of both primary and secondary school children. The authors found that most of the studies of universal free school lunch provision reported positive associations with quality of diet, food security and attainment. Improvements to family finances were also noted, particularly amongst low-income families [13]. 

In England, FSM are provided to all children up to and including year 2 of primary school (6–7 years of age). Older children are eligible for FSM provision only if their household is in receipt of certain benefits. Even then, annual household income must be below GBP 16,190 for those in receipt of child tax credit and below GBP 7400 for those in receipt of Universal Credit to qualify [14]. This is problematic as current thresholds mean that only one in three children living in poverty are eligible for FSM [15]. UFSM has never been implemented in UK secondary schools. 

To address rising levels of food insecurity in the borough, Hammersmith and Fulham, a local authority in London, have been implementing UFSM in two of their secondary schools since January 2020 with the aim of ensuring that no child is too hungry to learn. 

The aim of this study was to evaluate a pilot of Universal Free School Meals (UFSM) provision to children in two London secondary schools. In this paper, we explore the impact of the UFSM pilot on students and their families, through addressing the following research questions:

RQ1: What is the perceived impact of UFSM on students, including hunger, behaviour, and food consumption?

RQ2: What is the perceived impact of UFSM on family finance and food security?

Findings related to acceptability, feasibility and cost-effectiveness of the pilot will be reported elsewhere.

## 2. Materials and Methods

### 2.1. Study Design and Context

The study adopted a mixed-methods, quasi-experimental design. Two schools in which the UFSMs were made available were compared with two similar schools in the same local authority. Data were collected via a quantitative student survey; observations carried out by students at the intervention schools; and qualitative interviews with parents, staff and students. 

The setting was a borough in central London. The UFSM pilot began just ten weeks prior to the UK entering its first lockdown because of the COVID-19 pandemic. As such, schools were closed to most pupils between 18th March to early September 2020, and again between December 2020 and March 2021, meaning most students did not have access to UFSM during this time. In the periods that schools were open, there was additional disruption because of staff and student absences and COVID-19 mitigation measures. These disruptions, alongside data collection points, are shown in Figure 1. 

The intervention schools were chosen and recruited for inclusion in the pilot by the local authority, based on their relatively high level of need and their capacity to deliver the pilot. The two comparison schools that matched the intervention schools on key demographics (ethnicity, proportion of pupils eligible for free school meals, catering company, special education needs or not) were recruited by contacting the head teachers. Demographics for the two intervention and two comparison schools are shown in Table 1.

### 2.2. Description of the Intervention

In the intervention schools, all students were provided with a free school meal at lunchtime, comprising a main course and a pudding. In the comparator schools, lunchtimes continued as normal with free meals only being provided to those eligible under the existing, national FSM provision. The evaluation study began in March 2021 (Figure 1).

### 2.3. Student Survey

All students at the four schools were invited to complete an online survey during school hours. As the pilot was underway when the study began, it was not possible to administer a baseline survey, but we were able to look for signs of intervention impact by comparing the intervention and comparator group responses. Parents/carers and students were made aware of the survey in advance and advised that consent would operate on an opt-out basis, with an online opt-out form set up for parents (children additionally gave informed consent before completing the survey). The survey was set up online using JISC Online Surveys and took students around fifteen minutes to complete. It included questions related to household food security from the USDA (United States Department of Agriculture) Food Security Survey Module [16] and the consumption of high fat food and snacks using questions from the Habits Study [17] (Appendix A). All schools were offered a payment of GBP 1000 each for their participation in the study. 

Data analyses took place using Stata Version 17.0 [18]. For the high fat food and snacks items, we created a binary variable, in which the responses more than every day/every day/most days were grouped as high consumption, and once or twice a week/less than once a week/never grouped as low consumption of the item. We present proportions of students in each study arm who were high in consumption for each item. 

For hunger and food insecurity, we ran univariable logistic regression analyses with study arm as the exposure variable (intervention vs. comparator). Binary outcome variables were generated for hunger and food insecurity. The food insecurity measure was coded as 0 = secure and 1 = insecure. This was coded according to the USDA Food Security Survey Module guidelines [16], where raw scores of 0/1 are considered secure and scores of >1 are suggestive of food insecurity. The hunger variable was derived from a single question on the USDA Food Security Survey Module, which asked “in the last month, were you hungry but didn’t eat because your family didn’t have enough food?”. Responses to this question were either “Never”, “Sometimes” or “A lot”. From these, a binary variable of hunger was derived with “Never” responses coded as ‘0’ and “Sometimes” or “A lot” responses coded as ‘1′.

Both models were adjusted for gender as there was a gender bias in the data (75.2% male in the intervention schools compared to 16.2% in the comparator schools); (female = 0, male = 1). Clustering at the school level was accounted for using the command ‘melogit’ in Stata. Considering the quasi-experimental design and the small sample size, differences in outcomes between intervention and comparator schools, including 95% confidence intervals, are reported but no *p*-values are presented. 

### 2.4. Lunchtime Observations

Due to COVID-19 restrictions, the study team were unable to directly observe lunchtimes in the intervention schools. We therefore trained members of the student councils at both schools to collect these data. Students and their parents gave written, informed consent prior to participation. At each school, two online co-production sessions were held with the students to identify student priorities in relation to meal provision at lunchtime and develop observation sheets. Areas of focus suggested by students included queuing time, quality of food and availability of condiments. Students then worked in groups to complete observations over the course of one school week. At School 1, photographs of the food and canteen were also taken. As a result of discussions with students and staff at School 2 (the school for students with special educational needs), it was agreed that these students would not take photographs to simplify data collection procedures.

### 2.5. Qualitative Interviews

Parent/carers from the intervention schools were alerted to the study via participant information sheets, circulated through the schools’ parent email system. Interested parents were asked to fill out a short, online questionnaire about their child’s ethnicity, free-school meal status prior to the pilot, and food security status. The food security questions were taken directly from the USDA Food Security Survey Module [16], discussed above. Parents were classified, according to these responses, as either having high, marginal, low or very low food security. We sampled purposively with the aim of including parents with a range of experiences of household food security, FSM eligibility and child ethnicity. 

Staff participants were drawn from a purposive sample after the headteacher at each school informed staff with knowledge of lunch times of the opportunity to contribute. This included members of the leadership and catering teams at each intervention school. We also contacted all of the non-intervention schools in the borough to invite one member of staff for an interview via headteachers. This allowed us to explore the implementation of the national FSM scheme, and the potential changes that would be introduced by UFSM, with a wider sample of school stakeholders than just those in the two intervention schools. 

A sample of students at both intervention schools was recruited by the leadership teams based on their knowledge of students with a range of food insecurity experiences from across year groups. Staff distributed information sheets and consent forms to those students identified who expressed an interest in participating. Completed forms were returned to the study team. All student interviews at School 1 were paired and those at School 2 were individual. This was determined by the schools, based on the challenges of completing the interviews during a time of high COVID-19 transmission, available space, and large numbers of staff and student absences. The interviewer encouraged both students to answer questions individually during paired interviews and emphasised that both shared and differing views were acceptable.

Parent/carer and catering staff interview participants were offered a GBP 30 shopping voucher and students a GBP 15 shopping voucher for their participation. Schools were offered GBP 100 per staff interview in recognition of the time they gave. 

Due to COVID-19 restrictions, qualitative data collection took place either online or via telephone (according to participant preference). The interviews were conducted by one of two experienced, qualitative researchers (PEJ, VRC). At the start of each interview, participants were reminded of the aims of the research and verbal consent checked. Discussions were steered by the topic guides, which were used flexibly to ensure that key topics were discussed, whilst allowing for other important issues to arise. Interviews were audio recorded using encrypted audio devices and transcribed verbatim, by a University of Bristol Approved transcription service.

Data were analysed using the Framework Method [19,20], a type of thematic analysis. This method is particularly well-suited to team analysis, and the pragmatic demands of this study, which included tight timelines, and a mixture of inductive and deductive coding of the interview transcripts. The analytic process is shown in Figure 2. After anonymisation and data familiarisation, frameworks (one for each participant type) were drafted in Excel and tested by three members of the research team on a sub-set of transcripts, with the frameworks refined as necessary. The resulting codes were then applied to each transcript. Following this, data (both quotes and summaries against each participant under each theme) were manually charted into Excel before moving up the analytical hierarchy, to explore associations and patterns between the themes. 

All adult participants gave informed, online consent prior to interviews and students and their parents provided informed written consent prior to their interviews taking place. Ethical approval for the study was granted by the University of Bristol’s School for Social Policy’s Research Ethics Committee (SPSREC/20-21/151).

### 2.6. Integration of Qualitative and Quantitative Findings

We used a convergent design to integrate and present qualitative and quantitative findings [21]. Once we had analysed data from all sources (survey, observations and interviews), findings from each source were drawn on as appropriate to answer the two research questions.

## 3. Results

We carried out qualitative interviews with 60 participants, 20 parent/carers, 28 students (nine of which were paired interviews) and 12 staff. The mean interview duration was 41 min for parents (range = 20–68), 31 min for school staff (range = 17–61) and 28 min for students (range = 12–40). Characteristics of interview participants are shown in Table 2.

Additionally, at School 1, 39 observations were carried out by 13 students and at School 2, 20 observations were carried out by 11 students. A total of 408 students completed the survey (32.5% of all students), of which 404 complete responses were analysed. Characteristics of the survey sample are shown in Table 3. 

We developed five themes, which reflected findings in relation to the impacts of UFSM on students and their families. Together, these encompassed the following domains: (1) hunger/nutrition; (2) family finances and food insecurity; (3) educational impacts; (4) stigma/shame; and (5) social and emotional impacts. 

### 3.1. Hunger and Nutritional Impacts

The findings of the student survey are shown in Table 4 (findings regarding food insecurity discussed in Section 3.2). The survey did not show a clear effect of the intervention on hunger, with 13.1% of students at the intervention schools and 11.6% of students at comparator schools reporting feeling hungry ‘sometimes’ or ‘a lot’ in the previous month (see Table 4). Reports of hunger were slightly higher in the intervention schools than comparator schools; (OR = 1.09; CI = 0.29–4.11) with boys reporting being hungry (13.37%) more than girls (9.85%) (OR = 1.11; CI = 0.47–2.67).

Although none of the students interviewed reported not having enough to eat, they were aware that some of their peers were previously going hungry at school and were pleased that this was no longer the case: “I would say that more people have the opportunity to have lunch and they are able to go through the day without being hungry” (Year 10 student). The perception from all groups interviewed was that many students skipped breakfast. This was felt to be due to a lack of time or students not feeling like eating early in the morning, which sometimes resulted in students feeling hungry in the mornings. UFSM provision was perceived to be particularly important for these students, some of whom would otherwise be eating a less nutritious packed lunch or skipping lunch. Students expressed concern that the impacts of school meals on student hunger would be reversed if UFSM were discontinued. 

Table 5 shows the results of the student survey questions about food consumption in the intervention and comparator schools. Across all food and drink types, students at the intervention schools reported consuming unhealthy items more frequently than those at the comparator schools. For instance, 33 percent of students at the intervention schools reported eating crisps more than once a week, compared to 18 percent at the comparator schools.

Parents felt that it was important for students to have a hot meal at lunch time, particularly in the winter months; school meals were perceived to be more substantial, varied and satisfying than a packed lunch. Some parents felt that their children were eating a wider variety of foods at home because of being introduced to new foods via the pilot. 

The nutritional benefits of school meals were felt by staff to be particularly important for low-income families, whose children may be skipping meals, or only have access to poor quality food at home:

‘Cause every child comes in now to have a lunch whereas before they wouldn’t which I think is a good thing ‘cause obviously some kids might not eat a lot at home. So I just think it’s nice to be able to offer that free school meal for every student.’ (Catering Staff, Intervention School 1).

School meals were generally perceived to be healthy by parents, students and staff; however, a minority of parents (particularly at School 2) were unhappy with the quality of meals. The nutritional benefits of school meals may be undermined by student choice and preference. For instance, the photographs that the students at School 1 took as part of their observations showed a broad range of foods available at the counter including salads and vegetables. The photographs of the students’ plates, however, showed a noticeable absence of fresh fruit and vegetables and mostly comprising carbohydrates and cheese: “Even if they [canteen] have salad or something, people don’t take it. They just take the junk food, and they mostly have cakes and custard which has a lot of sugar and is not healthy”. (student, Year 10, School 1). This finding has important implications for food waste as well as student nutrition. 

Both schools offered a pudding as part of UFSM provision, and the student observation data suggest that this was typically high in sugar and low in essential nutrients. Similarly, the student observations and interviews suggest that the food available for students to buy in School 1 at break-times was also low in nutrient density, for instance, sweet waffles or pizza. Conversely, UFSM was perceived by some to have a positive impact on snacking behaviour via reduced snacking after school and removing the need to bring money, which students said was previously sometimes spent in the local shops on junk food, rather than a school meal: 

“Going off and having you know lunch of their choice, most of them would go to the chicken shop or just buy things from the corner shop. That hasn’t happened now since we started this project” (Leadership Staff, Intervention School 1). 

Some parents also observed that their children appeared to be less hungry when returning from school and were snacking less at home.

### 3.2. Impacts on Family Finances and Food Insecurity

The findings of the student survey showed that 29.6% of students at the intervention schools and 25.8% of students at the comparator schools reported some degree of food insecurity within the previous month (see Table 4). There was no clear evidence of an effect of the intervention on family food security (OR:0.93; CI = 0.25–3.52). The interview data, however, revealed that the intervention was perceived to have a beneficial impact on household food security. 

The financial benefits of UFSM were perceived to be cumulative, particularly for families with multiple children:

‘But the long-term impact of parents not having to fund school lunches for three years, I think is a really big thing and most of our families are low income so it’s—for me that was the thing I could see being really beneficial.’ (Leadership Staff, Intervention School 2). 

At School 1, prior to UFSM, children were able to spend as much or as little as they chose on school meals based on what their parents added to their payment cards; however, the cost of a set school meal (main course and pudding) to families in both intervention schools was previously GBP 2.30 per head. This has resulted in a direct saving to families of GBP 11.50 per school week for families with one child at an intervention school, and parents and students reported that this has freed up funds that can either be spent on food at home or on other enrichment activities. This was seen as being particularly valuable for those parents who were previously ineligible for FSM despite struggling financially. 

Whilst means-tested FSM was seen as useful in supporting families in need, there were concerns that the income threshold for eligibility was too low and was failing to support enough families “[…] I would say there’s then another sort of line of students who aren’t eligible but are still very deprived.” (Leadership Staff, Non-intervention School). 

Although the survey suggests that food insecurity amongst students is concerningly high already, this may in fact be an underestimation. Data from the parent/carer interviews suggest that parents conceal the realities of household food insecurity from their children by skipping or reducing the size of their own meals to ensure their children can eat, as this parent (previously FSM eligible) describes:

‘As much as it is hard, and it can be really hard at times I’m forever grateful that I’m able to make sure it doesn’t affect [son], you know and try and be the best mum I can and make sure that even if I am struggling, he doesn’t know, you know?’ 

The students we interviewed were aware of the financial challenges facing families at this time; however, most spoke about experiences of friends, rather than their own families. They felt that UFSM had been very beneficial for these families. 

### 3.3. Educational Impacts 

The qualitative data, particularly from students and staff, suggested that UFSM is perceived to be associated with a range of improved education-related outcomes including concentration, behaviour, and energy levels. As one student said, “if you eat well you will get to concentrate in lessons”. There was perceived to be a relationship between eating well and concentration, particularly after lunch. Having enough to eat was also perceived as being important for energy levels and mood:

‘It gives you a bit of energy doesn’t it so it can really, really impact them. It can make them feel really low, it can make them feel grumpy if they don’t eat enough, especially if they’re not getting the right food, it can really affect their attention and everything. Like it changes their whole mood. Once they eat, you can see the change in some of them. Gives them a burst of energy after lunch.’ (Teaching Staff, Intervention School 2). 

One member of staff reported that, even if behaviour and concentration did decline in the afternoons, they can now be confident that hunger is not the underlying cause, which allows them to focus on other strategies for managing classroom behaviour.

### 3.4. Impacts on Stigma and Shame

Most of the interviewees perceived stigma in relation to FSM had been reduced by widespread FSM eligibility at the schools and the use of pre-paid cards prior to the introduction of UFSM. However, the feelings of shame associated with claiming FSM was still regarded as problematic:

‘[…] you know you’re a child who’s receiving a handout and that can’t but help to have some kind of psychological impact on young people […] It’s something that we can so easily remove from being a worry for them. […] This concept that you had to beg is perhaps going too far, but that you had to fill in something to show that you’re so poor that the state’s going to take care of your lunch for you. I mean, it just can’t help.’ (Teaching Staff, Intervention School 1). 

Feelings of shame and difference may be felt more acutely by young people, who are particularly attuned to their social worlds at this stage of their development, as one parent (not previously eligible for FSM) noted: 

‘The problem is, if one kid hasn’t got any money on the card because him mum or dad can’t afford it, then it becomes noticeable to the children around you—children notice everything. You think they don’t, but they notice [….].’

One parent discussed that her son (previously eligible for FSM) had always refused school meals in the past due to his fear of being bullied for receiving them. She was happy to report that he now had a school meal every day and felt that his social skills had improved as a result. Together, these findings suggest that whilst enacted stigma towards FSM may have been partially addressed, UFSM may be effective in reducing the internalised and anticipated stigma that remains. 

### 3.5. Social and Emotional Impacts

After financial impacts (discussed below), parents saw the social outcomes as the next most important benefit of UFSM, with several parents feeling that their child’s social skills had improved because of eating each day with their peers. As one parent noted, “children need to be around children”. 

Students reported that they enjoyed eating with their friends, noting that some of their peers would miss out on the social aspects of eating if they skipped lunch before the pilot. Another perceived social benefit was the removal of the need to top-up lunch cards, which sometimes resulted in students feeling pressured to buy food for friends without money before the pilot was introduced:

‘Some people might ask me for money, if I can buy them something. Sometimes I don’t have money but if I have money, I will buy it for sure for them. When they changed the lunch to free, everyone had a chance to get food. Everyone had to eat, not go home hungry.’ (Year 10 student, School 1).

For parents, knowing that their child will receive a substantial meal whilst at school meant “one less thing to worry about” in their busy and stressful lives. As this parent (previously eligible for FSM) acknowledged, parental stress impacts on children’s wellbeing: “If the parent is not under pressure the children are happier”. Many of the parents talked about UFSM providing them with “peace of mind”, with UFSM being just one aspect of their child being nurtured in the school environment: 

“He’s learning there. He’s safe. He’s warm. He eats. And it’s like he comes home at 3:30 better off than he was in the morning. That’s like a big peace of mind for parents.” (Parent/carer, not previously eligible for FSM). 

Many of the parents interviewed were employed, and they appreciated the benefits of no longer needing to prepare a packed lunch or remember to give their child money for lunch in the mornings. This also benefited students, many of whom said they made their own packed lunch before

‘You don’t need to wake up near 6am and 7am making a packed lunch.’ (Year 9 student, School 2).

School staff reported that some families struggle to apply for FSM even when eligible. This may be due to having English as a second language, failing to remember to re-apply each year as well as fear of stigmatisation (although the latter was not perceived to be widespread). Staff felt that the switch to UFSM had removed these concerns and ensured that nobody misses out on a school meal. Parents also felt that the previous FSM system was difficult to navigate and described the wait to find out if eligible for FSM and having to manage in the meantime as stressful. 

## 4. Discussion

The findings from our evaluation of UFSM in two London secondary schools are mixed. The qualitative data do indicate a number of potential benefits. UFSM was perceived as an effective way to address student hunger and household food insecurity, to reduce stigma among those receiving free school meals, and to have a positive impact on educational, social and behavioural outcomes, and parental stress. However, the quantitative survey found no evidence of effect on student hunger, improved nutrition or family food insecurity.

Survey data showed that students at intervention schools were eating unhealthy food more frequently than peers in comparator schools. The provision of puddings could be one explanation for this, as well as the ability of students to purchase unhealthy snacks at School 1 at breaktimes. In contrast, other research has shown that packed lunches tend to be less healthy, containing higher proportions of ultra-processed foods (UPF) than school meals [22]. Another potential explanation for our findings in relation to the consumption of unhealthy foods is the gender bias in the survey responses, due to a higher proportion of boys attending the intervention schools. Gender is known to impact on food choice with girls consuming less UPFs and eating more fruit and vegetables than boys [22,23,24]. We also found an apparent gap between what is offered and what is consumed by students in the pilot. For instance, vegetables and salads are provided but students indicated that these are not often eaten. Choice architecture (‘nudging’) interventions, such as altering the placement of food to influence food choice, have shown some promise in improving the uptake of healthy food in schools [25]; however, these may be less effective in increasing vegetable consumption [26]. 

The study was conducted during the COVID-19 pandemic and the ‘cost of living crisis’ in the UK, where one-in-six UK households are now experiencing serious financial difficulties [27]. Our qualitative findings reflected the national picture, with most of the parents interviewed describing increasing pressures on family finances, not just among those previously eligible for FSM. It is estimated that 37% of school aged children living in poverty in England are not currently eligible for FSM; this compares to just 17% in Scotland where FSM are available to all primary aged children [28], suggesting that the current system may be failing families [29]. Indeed, our qualitative data showed that UFSM was perceived to be particularly beneficial for those families that were feeling financial pressures but did not meet the threshold for means-tested FSM, as it provided a valuable safety-net.

Some of the parents we interviewed described their experiences of food insecurity and the ways that they manage it, for instance, by skipping meals or eating less themselves to ensure their children had enough to eat. These experiences were stressful and worrying for parents, and this stress was seen to impact on the whole family. UFSM has the potential to positively impact on mental health and quality of life as addressing food insecurity can lead to an improvement in these outcomes [6]. Parents also described how they shielded their children from these realities, something observed in previous studies [30,31,32,33], suggesting that food insecurity scores from our student survey are likely to be an underestimate. It is known that even marginal levels of food insecurity are associated with poorer educational and behavioural outcomes [7], and other studies show that this may be improved with provision of UFSM [11,13]. Our qualitative findings support this, showing perceived improvements to both of these outcomes in the intervention schools as a result of UFSM. Like others [28,29], we argue that the expansion of FSM is necessary to support the many families now experiencing food insecurity in this country.

Another perceived benefit of UFSM is the elimination of stigma associated with receiving means-tested FSM. The issue of stigma that surrounds FSM is well documented [29,34,35,36]; whilst it may be possible to limit stigma towards FSM, e.g., by adopting a pre-payment card system, there are occasions where it is not possible to shield the identity of those receiving FSM, for instance, on school trips. In England, the take up of FSM by those eligible is 89% and one possible reason for those not making use of school meals is stigma and shame associated with doing so [34,37]. In some schools, students receiving FSM are restricted to purchasing from a limited menu, are unable to sit with friends or do not have enough funds to purchase a snack at break time as well as a meal at lunchtime [35]. In the current study, snacks were available for purchase at breaktimes at one intervention school, potentially creating a two-tier system whereby only wealthier students could afford snacks, thereby potentially perpetuating stigma. 

Our study had a number of strengths. To our knowledge, this is the first evaluation to be conducted of UFSM in a UK secondary school context. Previous studies have largely focussed on universal breakfast provision or UFSM in primary schools [13] and our findings extend this work to secondary schools. By collecting qualitative data from a large number of multiple informants, we have built upon the previous literature to add a more nuanced understanding of the many benefits associated with UFSM. In addition to this, our quantitative data add important contextual information about the high levels of food insecurity among our population. 

However, the study, in particular the quantitative element, also has limitations. The fact that only two schools were receiving the intervention, and the challenges of collecting data from schools during the COVID-19 pandemic, meant the survey sample was small and response rates were low. As a result, the survey was likely to be underpowered, and may have been subject to bias. Further, the non-randomised selection of schools meant there was a gender imbalance between the intervention and comparator groups, and the lack of baseline data limited our ability to detect any differences in outcomes that may have arisen due to the intervention, which may explain the discrepancy between our quantitative and qualitative findings regarding the impact of UFSM. Dietary impacts were only measured using a self-report measure of the consumption of high fat food and snacks.

Our qualitative sample had to be recruited by school staff as we were unable to visit the schools and this may have resulted in a sample more favourable towards UFSM, although we note we achieved good variation in the levels of food security experienced by parent participants. It is not known which of the student respondents were eligible for FSM. Though the interviewer tried to encourage individual responses during paired student interviews in School 1, this may have introduced some bias in terms of socially desirable responses.

## 5. Conclusions

Our research provides novel and timely evidence that, in the context of a worsening ‘cost of living crisis’ in this country, UFSM may be one potential strategy to address family food insecurity, student hunger, and to improve access to healthier foods at a key time of physiological and cognitive development for young people. This evidence comes largely from the qualitative data collected from students, parents and school staff, and may not be generalisable. The quantitative survey data showed no evidence of positive impact on student hunger, improved nutrition or family food insecurity. Future research should test out UFSM with a larger and more diverse sample of schools and collect baseline and follow up measures from intervention and comparison groups, to examine quantifiable long-term impacts of UFSM on health, financial and educational outcomes. It is also important to evaluate the cost effectiveness of such an approach, compared to other possible interventions to address food insecurity. 

## Figures and Tables

**Figure 1 ijerph-20-05216-f001:**
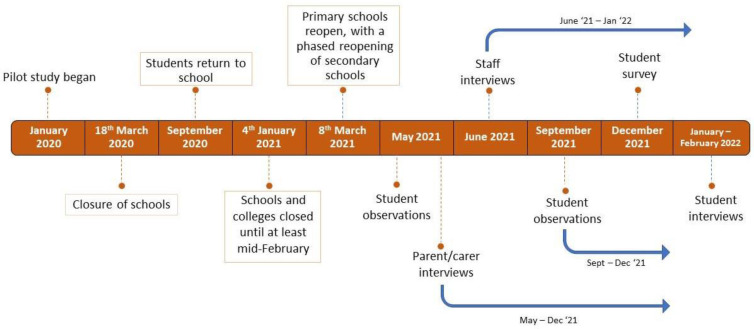
Timeline of data collection and key educational COVID milestones.

**Figure 2 ijerph-20-05216-f002:**
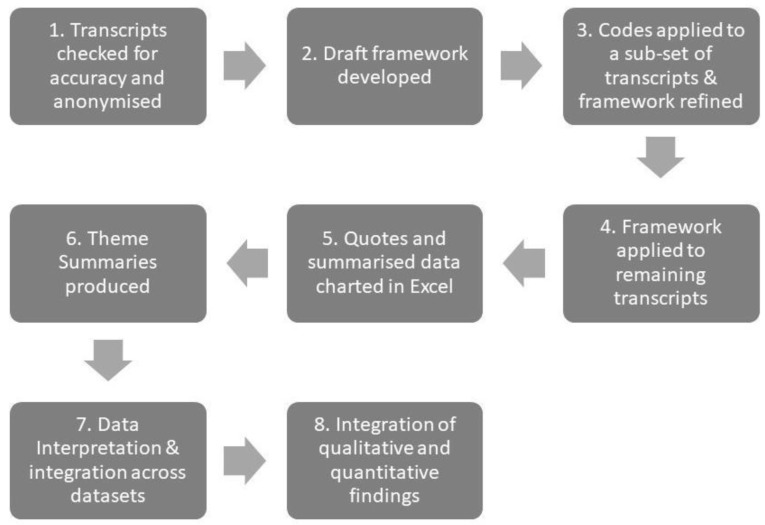
Qualitative analysis process using the Framework Method.

**Table 1 ijerph-20-05216-t001:** Characteristics of Intervention and Comparator Schools.

School	Type	Population	No. of Pupils	FSM Eligibility (%)	% Boys	% Non-White
1	Intervention	Mainstream, mixed ^a^	414	33.9	77.8	77.1
2	Intervention	Special School, mixed	105	46.6	69.4	56.4
3	Comparison	Mainstream, girls’	619	31.9	0.6	77.7
4	Comparison	Special school, mixed	117	68.4	75.6	56.1

Note: FSM = free school meals (means-tested) as proportion of school population. ^a^ Boys’ school until recently.

**Table 2 ijerph-20-05216-t002:** Characteristics of interview participants (*n* = 60).

	School 1	School 2	Non-Intervention Schools ^a^
**Parent/carers (*n* = 20)**			
FSM eligibility			
Eligible	5	5	-
Ineligible	8	2	-
Food security			
Low/very low food security	4	6	-
High/marginal food security	9	1	-
Ethnicity of child			
White	5	2	-
Mixed	2	2	-
Asian	2	-	-
Black	2	3	-
Other	1	-	-
Undisclosed	1	-	-
**School Staff (*n* = 12)**			
Leadership Team	2	1	3
Teaching Staff	2	1	1
Catering Staff	1	1	-
**Students (*n* = 28) ^b^**			
Year 7 (11/12 years)	4	1	-
Year 8 (12/13 years)	4	3	-
Year 9 (13/14 years)	4	2	-
Year 10 (14/15 years)	4	2	-
Year 11 (15/16 years)	2	2	-

Note: FSM = free school meals (means-tested). ^a^ Three of these staff members worked at the comparator schools. ^b^ All interviews in school 1 were paired, all interviews in school 2 were individual.

**Table 3 ijerph-20-05216-t003:** Characteristics of survey participants (*n* = 404).

	Intervention *n* (%)			Comparator *n* (%)		
	School 1 (*n* = 146)	School 2 (*n* = 62)	Combined (*n* = 206)	School 3 (*n* = 157)	School 4 (*n* = 41)	Combined (*n* = 198)
**Gender**						
Boy	112 (77.8)	43 (69.4)	155 (75.2)	1 (0.6)	31 (75.6)	32 (16.2)
Girl	30 (20.8)	16 (25.8)	46 (22.3)	149 (94.9)	8 (19.5)	157 (79.3)
Prefer not to answer	1 (0.7)	2 (3.2)	3 (1.5)	3 (1.9)	1 (2.4)	4 (2.0)
Prefer to self-describe	1 (0.7)	1 (1.6)	2 (2.3)	4 (2.6)	1 (2.4)	5 (2.5)
**Year Group**						
7	52 (36.1)	15 (24.2)	67 (32.5)	40 (25.5)	-	40 (20.2)
8	46 (31.9)	13 (21.0)	59 (28.6)	18 (11.5)	7 (17.1)	15 (12.6)
9	4 (2.8)	14 (22.6)	18 (8.7)	42 (26.8)	6 (14.6)	48 (24.2)
10	4 (2.8)	10 (16.13)	14 (6.8)	3 (1.9)	13 (31.7)	16 (8.1)
11	38 (26.4)	10 (16.13)	48 (23.3)	54 (34.4)	15 (36.6)	69 (34.8)
**Ethnicity**						
White	33 (22.9)	27 (43.6)	60 (29.1)	35 (22.3)	18 (43.9)	53 (26.8)
Mixed	11 (7.6)	5 (8.1)	16 (7.8)	25 (15.9)	12 (29.3)	37 (18.6)
Asian	6 (4.2)	2 (3.2)	8 (3.9)	18 (11.5)	1 (2.4)	19 (9.6)
Black	42 (29.2)	15 (24.2)	57 (27.7)	39 (24.8)	6 (14.6)	45 (22.7)
Other	52 (36.1)	13 (20.1)	65 (31.6)	40 (25.5)	4 (9.8)	44 (22.2)

Note: White ethnicity includes UK, Irish, Gypsy and any other white background; Mixed ethnicity includes White and Black Caribbean, White and Black African, White and Asian and any other mixed ethnic background; Asian/Asian British ethnicity includes Indian, Pakistani, Bangladeshi, Chinese and any other Asian background; Black/Black British ethnicity includes African, Caribbean and any other Black background; Other includes Arab and any other ethnic group.

**Table 4 ijerph-20-05216-t004:** Results of the student survey showing univariable logistic regressions of the impact of UFSM on student hunger and food insecurity.

	Hunger		Logistic Regression	
	Never *n* (%)	Sometimes/ a Lot *n* (%)	OR ^a^	95% CI ^b^
**Type of school**				
Comparator	175 (88.38)	23 (11.62)	Ref ^c^	-
Intervention	179 (86.89)	27 (13.11)	1.09	0.29–4.11
**Gender**				
Female	183 (90.15)	20 (9.85)	Ref	-
Male	162 (86.63)	25 (13.37)	1.11	0.47–2.67
	**Food Insecurity**			
	Secure	Insecure	OR	95% CI
**Type of School**				
Comparator	147 (74.24)	51 (25.76)	Ref	-
Intervention	145 (70.39)	61 (29.61)	0.93	0.25–3.52
**Gender**				
Female	160 (78.82)	43 (21.2)	Ref	-
Male	125 (66.8)	62 (33.2)	1.43	0.72–2.83

Note: ^a^ OR = odds ratio. ^b^ 95% CI = 95% confidence interval. ^c^ Ref = reference group in logistic regression analysis.

**Table 5 ijerph-20-05216-t005:** Consumption of unhealthy foods and drinks (most days, every day or >every day).

	Type of School *n* (%)	
	Intervention (*n* = 206)	Comparator (*n* = 198)
Crisps	68 (33.01)	35 (17.68)
Sweets	76 (36.89)	58 (29.29)
Cakes	29 (14.08)	16 (8.08)
Other puddings	66 (32.04)	21 (16.2)
Biscuits	71 (34.5)	49 (24.75)
Chips	58 (28.16)	28 (14.14)
Sausages/burgers	40 (19.42)	22 (11.11)
Sugar-sweetened beverages	29 (14.08)	21 (10.61)

## Data Availability

Qualitative Data presented in the study have been made available (on request due to confidentiality and ethical arrangements) at the ‘data.bris’ repository at 10.5523/bris.3rdt9hl6g2in72krp33j9693w0. Other data are not publicly available; however, academic researchers may send data requests to the corresponding author for review, and subject to confidentiality and data-sharing agreements.

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
