# Peer review of "A Mixed Methods, Quasi-Experimental Evaluation Exploring the Impact of a Secondary School Universal Free School Meals Intervention Pilot"

_ijerph, 2023, doi:10.3390/ijerph20065216_

Round 1

Reviewer 1 Report

Overall, this is an interesting piece of research that is relevant to current debates regarding the extension of FSM in the UK. This study uses a mixed-methods quasi-experimental evaluation exploring the impact of a secondary school universal free school meals intervention pilot (using 2 intervention and 2 comparator schools). The study team have explored a range of factors related to the implementation of this ‘intervention’, and appreciate the evaluation of such interventions are complex. However, my main concerns relate to the lack of several major limitations. Whilst the methods provided are satisfactory the authors do not adequately acknowledge their limitations under study limitations. This is a major flaw and much more detail is required, considering the statements mentioned in the wider discussion section.

Comments are noted below for the authors to consider:

11. Line 17 add ‘the’ after on and before impact

22. Line 90-92 would be better moved closer to sections on student surveys, lunchtime observations and qualitative interviews, and provide more clarity that the student survey and lunchtime observations are the quantitative date.

33. Results: Table 2 on characteristics mentions 51 of 60. Numbers need clarification – did 8 not participate?  

44. Table 4: text surrounding this only discusses hunger and then later in the results it picks up the findings on food insecurity. Needs something to allude to the fact this will be considered under section x. 

55. Line 233: discusses intervention no clear effect on hunger, this should be re-positioned in discussion and then potential reasons why. In line 407 mentions had an effect on student hunger, needs to be consistent.

66. Line 347: there is a comment on stigma and use of pre-paid cards; it would need to be clarified if this was ‘new’ with the initiative as a payment method in schools or was already in place.

77. Conclusions drawn based on method for dietary data: whilst using adapted USDA FSM and Habits Study, one question posed on experiencing hunger over the last month; allocated to ‘a lot’ or ‘sometimes’ and the method used for food insecurity question – needs acknowledged the major issues with this for diet and wider conclusion

88. Paired interviews for data collection (line 179 mentions practicalities but no  mention re potential bias/impact on pupils response)

99. Describe the purpose of contacting non-intervention schools in borough, confusing in results section, and where this ‘fitted’ in analyses

110. Need to acknowledge not known if pupils were FSM or non-FSM eligible regarding interpretations and limitation of conclusion on dietary impact of hunger/food insecurity and no detailed data collected to collaborate qualitative quotes

111.  Qualitative quotes from pupils seem to only be from Y10 pupils (no identification re school type); few quotes referenced throughout. Similarly, leadership quotes (intervention schools but is this – 1 or 2)

112.  There is a quote/comment that staff can now be confident behaviour not related to hunger – again needs more caution in how you use this – there is no quant evidence to support this. Generalisability of findings need acknowledged in limitations

113.  Discussion has too many occurrences where statements are posed which  the findings of this study don’t provide robust evidence for. While extension of FSM has growing support, the discussion needs to relate to these findings. For example, authors mention ‘argue that the expansion of FSM is necessary’. More caution would be better that this is a potential solution to address issues of hunger and food poverty, but more robust evidence, larger sample, pre and post data etc would be required to consider effects. This is made clearer in conclusion, therefore statements throughout discussion need modified, along with, more detailed acknowledgement of limitations.

Author Response

Please see the atatchment

Reviewer 2 Report

I appreciate your study.   I think that many countries are considering universal food service meals.  This study can provide data to impact their decisions.

I do have concern that some schools were predominantly male or female, but overall you had a more uniform sampling.  

The article is clearly written and appreciate including the quotes as well as the quantitative information.   I think this makes the paper easy to read.  

I think you clearly explained while there are positive impacts of UFSM while not showing significance.

Author Response

Thank you for taking the time to review our article. We appreciate your kind comments. 

Reviewer 3 Report

The present manuscript is focused on the impact of a universal free school meals pilot in English secondary schools. The topic is really interesting because there is a growing concern regarding the food insecurity among the school students with lower income not only in England, but in many other parts of the world.

I have several remarks regarding the manuscript:

Materials and Methods

Authors should state how many students in the comparison schools have received FSM and how many of them have completed the survey.

“Other” ethnicity group is quite big in all schools and should be provided with description. “White” is also rather unclear in the frame of multiethnic profile of city of London.

It will be good if the authors provide a full version of the online survey, maybe as a supplement.

  Results

Comparison between the nutritional values and food variety between UFSM and FSM menus should be provided (Supplementary table or else). Are there specific standards for the nutritive values?

Student’s age should be presented in years so to allow universal comparability of the data.

Abbreviations in Table 4 are not presented, % as well.

Table 5:

Only consumption of unhealthy food is given. In the Introduction Authors claim to aim at “RQ1: What is the perceived impact of UFSM on students, including hunger, behavior, and food consumption?” Other foods should be added as well, especially vegetables and fruits that are commented in the qualitative results.

Most days, every day or >every day - should be presented separately.

Comparison between Special and Mainstream schools is missing.

Author Response

Please see the atatchment

Reviewer 4 Report

The United Kingdom is introducing many programs and scientific research concerning, for example, the influence of harmful junk food on the diet of people, especially children, and in efforts not only in the education of society but also practically introducing changes in children's food style of living. That paper concerns the benefits of free meals in two English secondary schools. It is a pilot study showing how difficult it is to overcome bad eating habits or food insecurity.

Tables and figures well document the text.

It is interesting to read the direct responses of student staff or parents.

For me, a very supporting, to easier comprise of many factors, would be a figure describing all aspects important in this project from the point of view of all involved (students, staff, parents).

The work is challenging to read due to the complexity of the problem. Therefore, the drawing indicating the factors included in the study at different levels can be a base for further work in this area by the publication's authors or others in implementing similar studies.

Round 2

Reviewer 1 Report

I would like to thank the authors for taking the time to address the comments provided in feedback - happy with their changes to the paper.